# Absence of reliable physiological signature of illusory body ownership revealed by fine-grained autonomic measurement during the rubber hand illusion

Hugo D. Critchley [1,2,3]*, Vanessa Botan[1,2], Jamie Ward[1,2]

**1** School of Psychology, University of Sussex, Brighton, United Kingdom, **2** Sackler Centre for Consciousness Science, University of Sussex, Brighton, United Kingdom, **3** Brighton and Sussex Medical School, University of Sussex and University of Brighton, Brighton, United Kingdom

* h.critchley@bsms.ac.uk

**Data Availability Statement:** All relevant data are within the paper and its Supporting Information files.

## Abstract

The neural representation of a 'biological self' is linked theoretically to the control of bodily physiology. In an influential model, selfhood relates to internal agency and higher-order interoceptive representation, inferred from the predicted impact of efferent autonomic nervous activity on afferent viscerosensory feedback. Here we tested if an altered representation of physical self (illusory embodiment of an artificial hand) is accompanied by sustained shifts in autonomic activity. Participants (N = 37) underwent procedures for induction of the rubber hand illusion (synchronous stroking of own unseen hand and observed stroking of artificial hand) and a control condition (asynchronous stroking). We recorded electrocardiography, electrodermal activity, and a non-invasive measure of multiunit skin sympathetic nerve activity (SKNA) from the chest. We compared these autonomic indices between task conditions, and between individuals who did and did not experience the illusion. Bayes factors quantified the strength of evidence for and against null hypotheses. Observed proprioceptive drift and subjective reports confirmed the efficacy of the synchronous (vs asynchronous) condition in inducing illusory hand ownership. Stringent discriminant analysis classified 24/37 individuals as experiencing the rubber hand illusion. Surprisingly, heart rate, heart rate variability, electrodermal activity, and SKNA measures revealed no autonomic differences between synchronous vs asynchronous conditions, nor between individuals who did or did not experience the rubber hand illusion. Bayes factors indicated substantial evidence for no physiological differences. In contrast to earlier reports, our autonomic data show the absence of a reliable change in physiological state during the rubber hand illusion. More encompassing perturbations of self-experience, for example in full body illusions, may nevertheless be coupled to, or facilitated by, changes in efferent autonomic activity and afferent viscerosensory feedback. Our findings suggest that such changes in bodily physiology are not sustained as an obligatory component of the rubber hand illusion.

**Funding:** This study was funded by a bursary from the Bial Foundation 128/14 to HC entitled: Microneurography and autonomic nerve recording as a tool for consciousness science.

**Competing interests:** The authors have declared that no competing interests exist.

## Introduction

The representation of the body's physiological state and physical boundaries is argued to be fundamental to self-perception and awareness. This representation is built upon the integration of sensory information across modalities with expectations and predictions regarding what our body should be doing [1–4]. The dependence of the experience of body ownership on dynamic coherence across senses is illustrated by the rubber hand illusion, induced through correspondence between somatosensory stimulation and visual feedback: An artificial 'rubber' hand is placed in front of a participant and seen to be stroked. At the same time, the matching own hand of the participant, is hidden from sight, yet stroked at the same frequency (synchronously). Most participants experience the rubber hand as part of their own body [4]. This experience is reported by the participant as a subjective feeling, potentially linked to internal (interoceptive) bodily sensations [5]. The strength of the illusion can be scored on questionnaires and visual analogue scales. Less subjective measures of the illusion include 'proprioceptive drift', where the judged location of the participant's own hand shifts to be nearer the rubber hand. Physiological reactions can also offer objective measurements of the rubber hand illusion. For example when the rubber hand is threatened, for instance with being hit by a hammer, the degree of illusory embodiment can be inferred from the magnitude of sympathetic skin conductance response [6] or from the strength of motoric withdrawal of the participant's own hand, evoked by the apparent threat [7]. However, even despite the relative consistency across studies of exaggerated reactivity to threats to an embodied rubber hand [8], these effects may be more nuanced, in some cases relating more to physical contact than threat [9,10].

Influential theories of consciousness suggest that there is a primacy to the (predictive) control of internal bodily state, through which the sense of self as a continuous, unitary, bounded experience arises from the inseparability of mental processes from the dynamics of interoceptive (inner physiological; viscerosensory) sensation and internal agency [11–14]. Empirical data to support this is more limited. However, abnormalities in normative autonomic reactivity or interoceptive processes are associated with clinical conditions involving disturbed self-representation [15,16]. Furthermore, individuals with reduced ability to perform a heartbeat tracking task (a heuristic measure of sensitivity to interoceptive signals) are more susceptible to the rubber hand illusion [5], suggesting that a weaker model of 'internal self' predisposes to a more malleable representation of the body's boundaries. Relatedly, if physiological information is introduced into the visual display of a virtual rubber hand or body (by colour changes pulsing normatively in time with the participant's own heartbeat), this can increase the likelihood (and objective correlates of) the illusory experience of body ownership, indicating a binding role of interoceptive predictive experience in the representation of the physical self [17,18].

Within active inference models of interoception, autonomic drive to peripheral organs encompasses predictions about the desired internal state, changing the afferent feedback to manage better interoceptive prediction error [13,14]. As noted, interoceptive predictions and internal agency are central to current theoretical models of embodied selfhood [13]. By extension, one can hypothesize that changes in peripheral physiological state, mediated autonomically, would reflect a shift in bodily self-representation associated with the experience of the rubber hand illusion. Such changes are distinct from measuring reactions to threat provocation, in that they are proximate to the putative interoceptive mechanism underpinning conscious selfhood. Relevantly, a fall in temperature of substituted (participant's own stroked) hand is described as an objective indicator of the illusion of embodiment [19]. Conversely, cooling the tested hand may facilitate the experience of the illusion [20]. In a stable

environment, cooling of the skin surface can occur through changes in skin blood flow (perfusion) and via increased sweat production (driven by sympathetic nerves innervating sweat glands, whose activity is reflected in electrodermal measures such as skin conductance responses). Immobility during the induction of the rubber hand illusion may underlie local autonomic changes. This passive consequence of proprioceptive quiescence is potentially amplified by the suggested/intentional state evoked by synchronous stroking. Here, the physiological (peripheral vascular) state of the participant's own hand better matches the cold lifeless visual appearance of the embodied rubber hand. However, not all studies have been able to replicate the finding of skin cooling in the participant's hand corresponding to the rubber hand [21,22].

Few studies have tested explicitly for cardiac changes associated with the illusory body ownership, despite some studies [5,17], but not all [23,24], demonstrating a relationship between 'cardioceptive sensitivity' (implying better heartbeat perception) and susceptibility to the rubber hand illusion, and other studies showing that the illusory experience can be modulated by heartbeat information [17,18]. Changes in heart rate and heart rate variability provide measures of the relative balance of antagonistic sympathetic and parasympathetic autonomic nervous activity on the atrioventricular pacemaker. This balance contributes, via the baroreflex, to blood pressure regulation. Stronger heartbeats increase afferent feedback from arterial aortic and carotid baroreceptors, which inhibits efferent sympathetic drive and enhances vagal parasympathetic slowing of heart rate. Cognitive, affective and behavioural states can engender an increase in cardiovascular arousal through 'top-down' suppression of the baroreflex, shifting the sympathovagal balance to increase heart rate and decrease (parasympathetically dominant) heart rate variability. Where studies have tested for changes in heart rate or heart rate variability in association with illusions of body ownership, few differences are reported [18]. In contrast, variance in electrodermal activity (not evoked by threat to the rubber hand illusion) may indicate the adoption of an illusory rubber hand within a participant's bodily representation, differentiating between the synchronous stroking condition (that best induces the illusion) versus asynchronous 'control' condition. However, these effects appear to be transient, suggesting that psychological novelty of the experience is a potential cause [25]. Changes in electrodermal activity, including tonic skin conductance level and phasic skin conductance responses, are an expression of sympathetic nerve activity regulating eccrine sweat gland function [26,27]. Electrodermal activity is usually measured from the palmar surface of the hand and has proved to be a sensitive and widely adopted index of central attention and arousal states. Of relevance to interoception, skin sweat glands do not possess dedicated visceral afferents, hence sensory feedback of electrodermal activity is absent or indirect (via correlated changes in physiological arousal in other organ systems; but see [28]). Moreover, reflecting organ specificity, the autonomic sympathetic innervation of the palmar skin is dissociable from sympathetic innervation of heart and vasculature [29].

Here, we tested if autonomically-mediated changes in bodily state would reflect the alteration in conscious bodily self-representation associated with adoption of the rubber hand illusion. Participants underwent a rubber hand illusion induction protocol [4,5,8–10]. This involved synchronous stroking of a visible artificial right hand and the participant's own hidden right hand. A control comparison condition was also conducted in which the stroking of the two (real and rubber) hands was asynchronous. We predicted that changes in body ownership (experience of the rubber hand illusion) would be apparent as differences in autonomic reactivity over synchronous compared to asynchronous stroking conditions and between those who did and did not experience the illusion. We recorded for analysis; 1) electrocardiography (to quantify heart rate and vagally-mediated heart rate variability, HRV); 2) electrodermal activity from the participants right hand (to quantify the magnitude and frequency of

sympathetic skin responses), and; 3) a relatively novel non-invasive proposed measure of skin sympathetic nerve activity (SKNA), recorded over the chest, in a distribution reflecting sympathetic activity of the stellate ganglion [30–32]. This SKNA measure was included as a fine-grained direct measure of efferent autonomic nerve traffic. We tested the hypothesis that cardiac, electrodermal, and SKNA autonomic indices would differentiate the synchronous induction vs asynchronous control conditions in the rubber hand illusion protocol, and between individuals who experienced the illusion to a greater extent [33]. Motivated by the theoretical and empirical link to perceptual aspects of interoception [5,13,14,17,18], we undertook exploratory analyses to test for associations between (objective and subjective) measures of the rubber hand illusion with individual differences in heartbeat tracking performance and interoceptive metacognitive insight (awareness) [34,35].

## Materials and methods

### Participants

A total of 37 participants (mean age = 21.59, SD = 3.69; 31 females) completed the study. Participants were recruited from students and staff at the University of Sussex via electronic advertisement. Ethical approval was obtained from the Brighton and Sussex Medical School Research Governance and Ethics Committee (BSMSREGC) and the Science and Technology Research Ethics Committee of the University of Sussex. All participants give written informed consent.

Sample size was computed using G-power calculator for paired t-tests. The computation parameters included a medium effect size (d = 0.5), α set at 0.05, and power set at 0.8. The resulting sample size was of 34 participants. Considering that the effect sizes between synchronous and asynchronous conditions are usually medium/large [19,36,37] a sample of 34 participants was considered adequate.

### Rubber hand illusion paradigm

Testing was conducted in a climate-controlled room. In the rubber hand illusion task, the participant's right arm was placed in a box (86 cm×60 cm×20 cm), screened from view. A life-size artificial 'rubber' model of a right hand was placed midline within the visible section of the box, directly in front of participant body. The distance between the participant's right index finger and the index finger of the rubber hand was 20 cm.

Stroking with a paintbrush was applied by the experimenter (VB) to the index finger of the participant's hidden hand and the artificial hand. Two conditions were performed in a counterbalanced order across participants: 1) Synchronous stroking when the timing of the brush strokes on the participant's own hand and on the rubber hand was coincided. 2) Asynchronous stroking when the timing of the brush strokes on the participant's hand and rubber hand was out of phase by approximately 625ms). Each condition lasted two minutes. At the beginning of each condition, the participant estimated the location of her/his right index fingertip three times by reading the corresponding number along a one-meter ruler, whose visible position varied each time to prevent the participant repeating responses in subsequent readings. Post-induction finger location judgements were obtained in the same manner as prior to the induction. Proprioceptive drift was calculated by subtracting the average of the pre-induction finger location judgements from the average of post-induction finger location judgement:

$$PD = mean(post-induction\ judgements) - mean(pre-induction\ judgements).$$

After each condition, the participant completed the RHI questionnaire comprising 10 items divided into three subscales: ownership, location, and agency, see *Table 1* for further details. The items were measured on a 7-point Likert scale (1 = strongly disagree, 7 = strongly agree) and the average score for each subscale was calculated [33].

## Physiological measures

**Heart rate and heart rate variability.** Electrocardiographic signals were recorded using CED 1902 and 1401 hardware and fed into Spike 2 software (Cambridge Electronic Design Ltd; Cambridge UK) applying a 10Hz high bandpass filter and 100 Hz low bandpass filter [38]. For the analysis, a threshold was applied to isolate R-wave peaks and to extract the number of heartbeats in a given time interval. This gave measures of heart rate (HR) (beats per time interval) and heart rate variability (HRV) expressed as the root mean square of successive differences (RMSSD) between normal heartbeats, the primary time-domain measure for short-term variation, strongly correlated with high-frequency variations and an indicator of the vagally-mediated (parasympathetic) changes reflected in HRV [39]. Both HR and HRV were calculated for synchronous and asynchronous conditions, averaged over the two minute for each condition.

**Electrodermal activity (skin conductance responses).** Electrodermal activity was recorded using two finger electrodes (CED2502) and electrolyte gel placed on the palmar surface of the index and middle finger of the participant's right hand. Signals were recorded into Spike 2 software using CED 1401 hardware. Analyses of skin conductance responses (SCRs) were performed on data exported to Matlab using Ledalab (V3.4.9) software. Adaptive data smoothing was applied, and continuous decomposition analysis was performed with extraction of continuous phasic and tonic activity. All SCR-onsets and amplitudes of above-threshold SCRs (a minimum of 0.01 microS) were then used to compute the average SCR amplitude per condition and a total number of SCR peaks per condition over the two minutes [26].

**Skin sympathetic nerve activity (SKNA).** Following methods described by Doytchinova and colleagues [30–32], six electrocardiographic electrodes were placed over the chest of the participant, two placed on the wrists, and two placed on the ankles (Fig 1). This signal was captured with PowerLab 16/35 (ADInstruments, Dunedin NZ) and recorded and displayed using LabChart v7.

To record SKNA, we followed methods described by Doytchinova and colleagues [30–32]: Electocardiographic (ECG) patch electrodes were used to record high frequency electrical

**Table 1. Rubber hand illusion questionnaire items and subscales.** (See [33]).

| Subscale | Items |
|---|---|
| Ownership | *It seemed like. . .* |
| | 1. . . .I was looking directly at my own hand, rather than at a rubber hand. |
| | 2. . . .the rubber hand began to resemble my real hand. |
| | 3. . . .the rubber hand belonged to me. |
| | 4. . . .the rubber hand was my hand. |
| | 5. . . .the rubber hand was part of my body. |
| Location | 6. . . .my hand was in the location where the rubber hand was. |
| | 7. . . .the rubber hand was in the location where my hand was. |
| | 8. . . .the sensation I felt was caused by the paintbrush touching (or laser pointer playing on) the rubber hand. |
| Agency | 9. . . .I could have moved the rubber hand if I had wanted. |
| | 10. . . .I was in control of the rubber hand. |

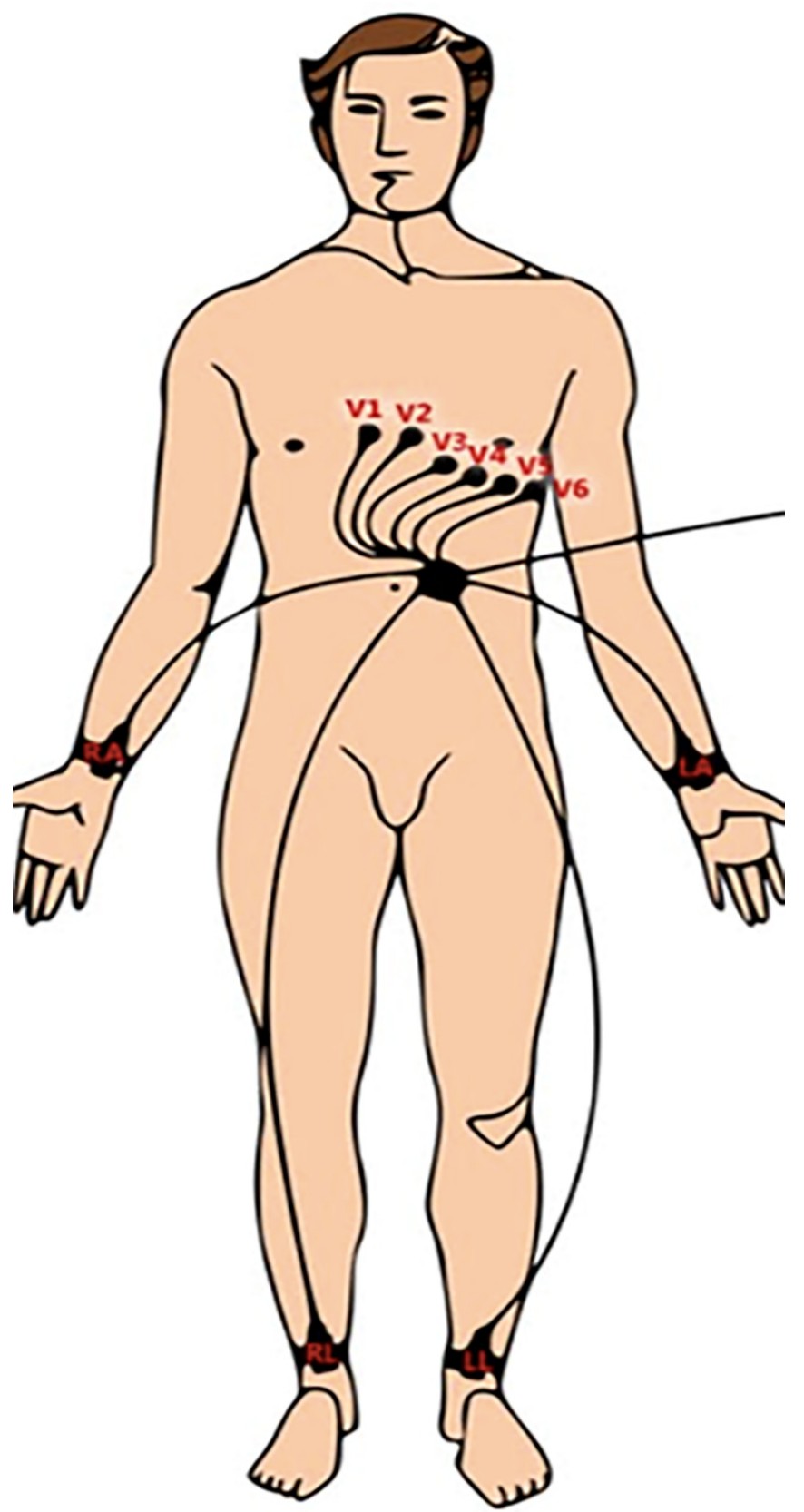

**Fig 1. Electrode positioning for recording of skin sympathetic nerve activity (SKNA).**

signals bandpass filtered between 500Hz and 1000Hz and derive the average signal (aSKNA) for each of the two experimental conditions. Correspondence between aSKNA activity and stellate ganglion function has been established in animal and human studies [32].

The sampling rate was set at 10000 samples/s and bandpass filtering between 500Hz and 1000Hz was applied. This filtering is shown to provide the best signal-to-noise ratio for skin sympathetic nerve activity (SKNA) recordings [30,32]. Quantitative signal data analysis was conducted, following the published protocol [30]. Here, the average SKNA (aSKNA) was calculated for specific time windows by dividing the total voltage within a time window to the number of samples within that specific time window at a sampling rate of 10,000/s. To test if SKNA would provide better temporal resolution than other autonomic measures, we initially used 30s-time windows; the total number of samples within the time window was 300,000. The 30s aSKNA was used in the initial statistical analyses presented below and further computed for 120s for comparison with physiological measures.

**Interoceptive accuracy and awareness.** Interoceptive accuracy was measured using the heartbeat tracking task [34,35] containing six trials with varying interval durations of 25, 30, 35, 40, 45 and 50 seconds played in a randomised order. The participant was instructed to silently count the number of heartbeats perceived in the given interval and to report them at the end of each trial. The actual number of heartbeats was measured using medical grade pulse oximeter with a soft sensor (Nonin Medical Inc, Plymouth MN USA; Xpod 8000S). For each trial, the accuracy score was derived using the following formula:

$$1 - (|nbeatsreal - nbeatsreported|)/((nbeatsreal + nbeatsreported)/2).$$

The resulting scores of each trial were averaged, yielding the overall value for each participant [35]. Confidence judgements were taken at the end of each trial, participants being asked to rate the confidence they had in their reported number of heartbeats. Their response was recorded on a 10 points continuous visual analogue scale (VAS) from 'total guess/no heartbeat awareness' to 'complete confidence/full perception of heartbeat'. Interoceptive awareness was then calculated using the Pearson correlation between interoceptive accuracy and confidence rating [35].

A median split was performed in exploratory analysis of the relationship between interoceptive performance on the heartbeat tracking task, The two groups (low and high interoception) had 17 participants each and the median point was 0.71. Due to technical issues, interoceptive data was not available for three participants (16 in the higher interoception group, 14 in the lower interoception group). Therefore, this reduced the degrees of freedom to 32.

**Statistical analyses.** Final statistical analyses were run in SPSS25 (IBM Statistics). Independent samples t-tests and their Bayesian equivalent (i.e. independent samples normal) tests were run on proprioceptive drift and subjective ratings for two groups (low vs high interoceptive accuracy) of seventeen participants each. Paired sample t-tests and their Bayesian equivalent (i.e. related samples normal) tests were performed to compare physiological measures between conditions. Pearson correlations were conducted across physiological measures. Pearson correlations and their Bayesian equivalents were also run to establish the relation between drift and interoceptive accuracy and awareness. Equal variances were assumed and Rouder's method was applied for the Bayesian analyses. When normality was not assumed (i.e. questionnaire data and aSKNA data), non-parametric tests were run to re-confirm the results (see Supplementary material).

A *discriminant analysis* was run to distinguish between participants who did and did not show a strong experience of the illusion, using the difference between synchronous and

asynchronous conditions in subjective ratings and in proprioceptive drift as predictors. The correlations between these predictors was relatively low (r = 0.367) and Wilks' Lambda was statistically significant for both predicting variables (p<0.001), confirming their adequacy for the analysis. Following this discriminant analysis, independent samples t-tests and their Bayesian equivalent (i.e. independent samples normal) tests were used to compare aSKNA results between the two groups of participants classified according to strength of their experience of the illusion.

## Results

### Behavioural results

**Proprioceptive drift.** Proprioceptive drift data was normally distributed for both synchronous and asynchronous drift measures (Shapiro Wilk normality tests; p = 0.091 and p = 0.945 respectively) and normality Q-Q plots. For the entire sample, proprioceptive drift was greater in the synchronous condition (M = 26.0 mm, SD = 5.46) than in the asynchronous condition (M = 4.6mm, SD = 3.67), t(36) = 3.097, p = 0.004, 95% CI [7.38, 35.41], d = 7.56, $B_{H(0,1)}$ = 0.123, thus there is moderate evidence for H1. Therefore, according to a typical control procedure, the RHI was successfully induced (Fig 2).

Each participant judged the location of their unseen index finger on prior to and after each synchronous and asynchronous stroking condition (synchrony relative to the observed

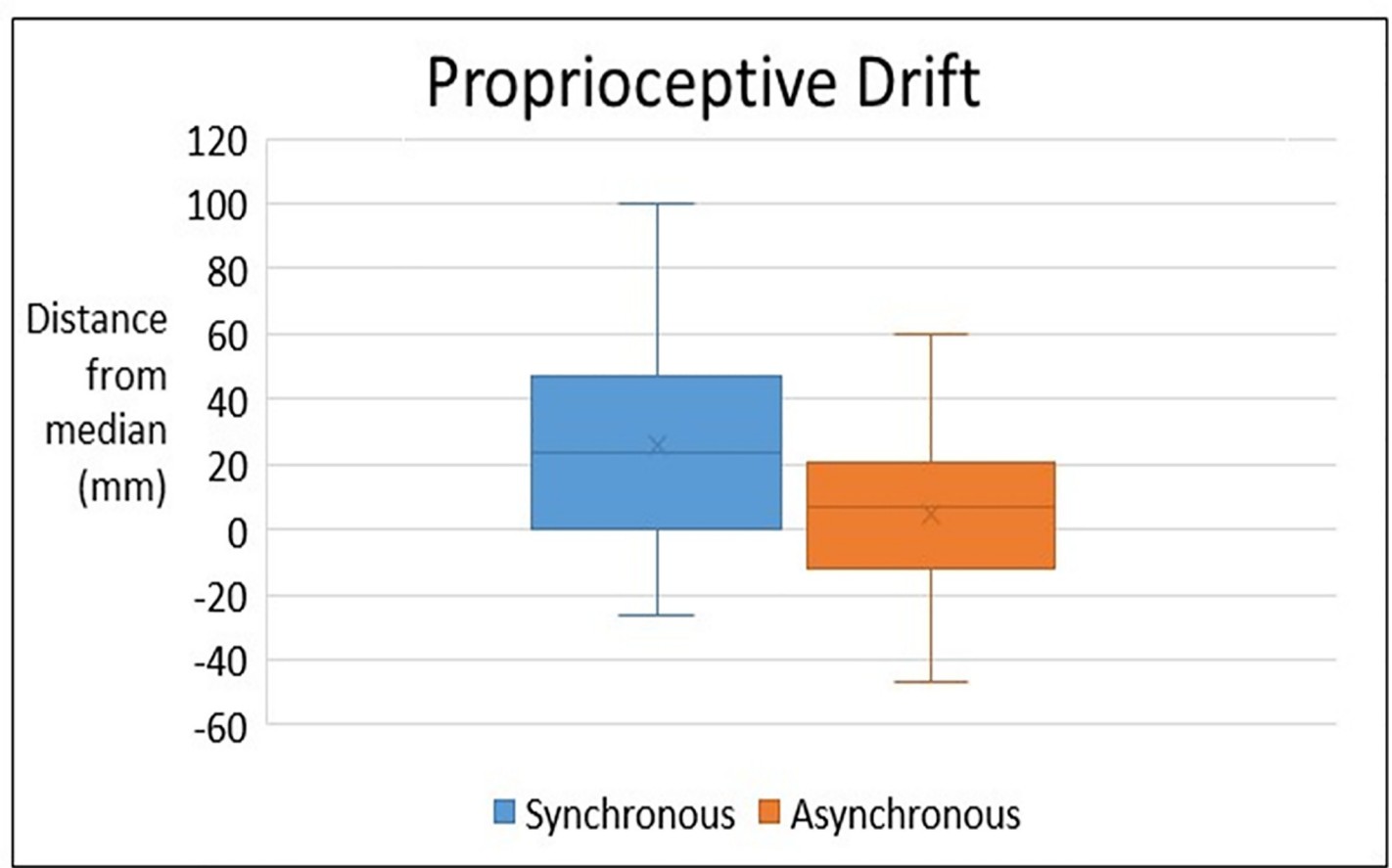

**Fig 2. Proprioceptive drift in synchronous and asynchronous conditions expressed in millimetres (mm: Median ± 95% CI).**

stroking of a visible artificial 'rubber' hand placed midline, directly in front of the participant). Proprioceptive drift is the objective finding that the experience of the illusion is accompanied by the participant judging his/her own hand to be located nearer to the artificial hand. The amount of measure 'drift' in a participant's rating of his/her unseen hand is thus used as an objective measure of the illusion. Here, this measure of embodiment shows the synchronous condition induced the illusory experience.

## Subjective reports

The three subscales of the rubber hand illusion questionnaire [33] were independently analysed providing the following results: ownership, $t(36) = 7.825$, $p<0.001$, 95% CI [1.23, 2.10], $d = 1.28$, $B_{H(0,1)} = 1.0 \times 10^{-3}$; location, $t(36) = 6.187$, $p<0.001$, 95% CI [0.87, 1.72], $d = 0.73$, $B_{H(0,1)} = 1.0 \times 10^{-3}$; agency, $t(36) = 5.036$ $p<0.001$, 95% CI [0.87, 2.00], $d = 0.85$, $B_{H(0,1)} = 1.0 \times 10^{-3}$. As such, there was decisive evidence for $H_1$, indicating that the illusion was successfully induced (Table 2; Fig 3).

**Discriminant analysis.** Participants with greater experience of the illusion could be distinguished by discriminant analysis from those with weaker or no experience of the illusion. There was a medium correlation between predictors (proprioceptive drift and subjective ratings) ($r = 0.367$) and Wilks' Lambda was statistically significant for both predicting variables ($p<0.001$), confirming their adequacy for the analysis. An Eigenvalue (= 2.220) explained 100% of the variance with a high canonical correlation of 0.830. The model indicated two groups (illusion vs non-illusion) and had a high sensitivity (91.7%) and specificity (100%). The discriminant analysis indicated that twenty-four participants experienced the illusion (21 Females, M = 22.15 yrs, SD = 4.81) and thirteen participants had minimal experience of the illusion (10 Females, M = 21.29 yrs, SD = 3.00). The groups did not differ by age ($t(35) = 0.763$, $p = 0.505$), nor gender ($\chi^2 = 0.694$, $p = 0.405$).

## Physiological results

**Heart rate and heart rate variability differences between synchronous and asynchronous conditions.** Across the entire group, over the two minute period of each condition, heart rate during synchronous rubber hand condition was shown (substantial evidence) to be equivalent to that observed in the asynchronous condition ($t(36) = 0.088$, $p = 0.930$, 95% CI [-1.84, 2.001], $d = 0.299$, $B_{H(0,1)} = 7.79$). Heart rate variability (RMSSD) during synchronous and asynchronous conditions was also similar (anecdotal evidence; $t(36) = 1.425$, $p = 0.163$, 95% CI [-2.41, 0.95], $d = 0.221$, $B_{H(0,1)} = 2.97$). Together, these results indicate that cardiac physiology 'sympathetic' heart rate and parasympathetic RMSSD did not differentiate the synchronous stroking condition, associated with the induction and experience of the rubber hand illusion from asynchronous 'illusion free' control condition.

**Heart rate and heart rate variability differences between participants who experienced the illusion and those who did not.** For the synchronous condition, we observed no

**Table 2. Showing means and standard deviations for subjective ratings in each condition and for each subscale of the rubber hand illusion questionnaire [33].**

| Self-rated experience 1 minimal to 7 maximal | Synchronous condition (mean ± S.D) | Asynchronous condition (mean ± S.D) |
|---|---|---|
| Ownership | 4.31 ± 1.95 | 2.65 ± 1.61 |
| Location | 4.14 ± 1.46 | 2.85 ± 1.33 |
| Agency | 3.53 ± 1.87 | 2.11 ± 1.38 |

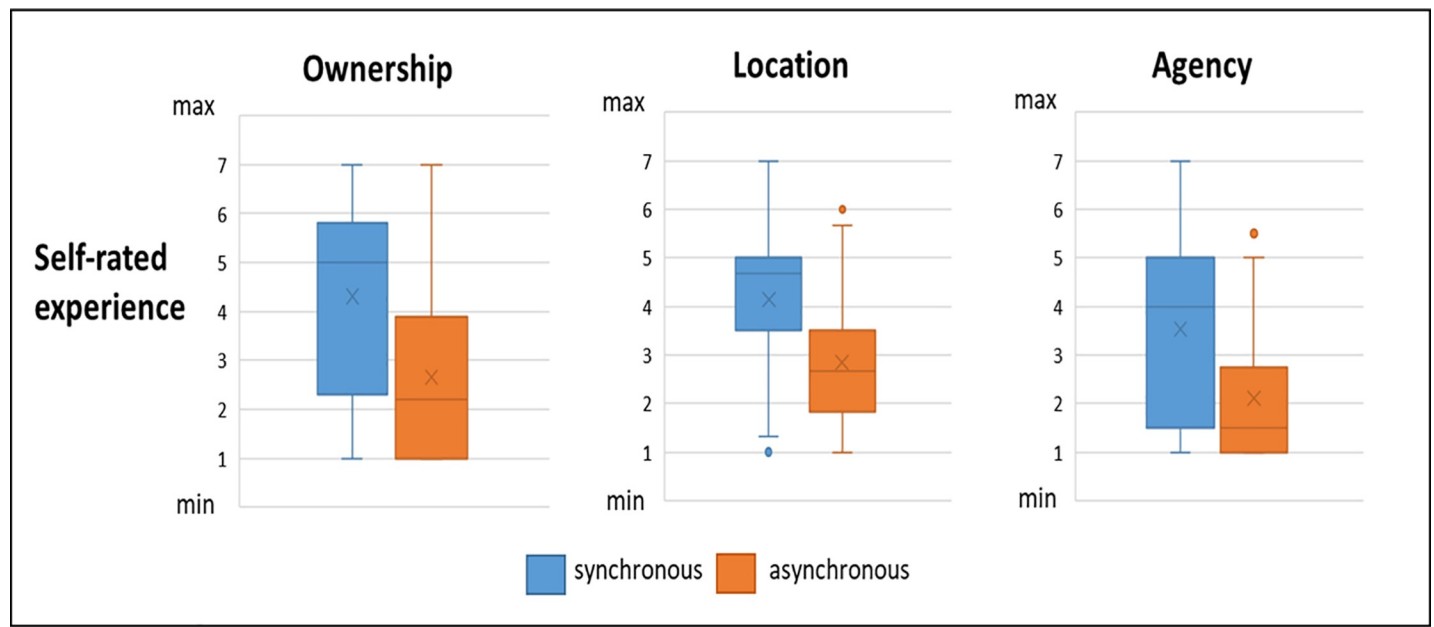

**Fig 3. Distribution of subjective ratings for each condition and each subscale of the rubber hand illusion questionnaire.**

differences in heart rate between participants who did and did not experience the rubber hand illusion (anecdotal evidence; $t(34) = -1.423$, $p = 0.164$, 95% CI [-12.86, 1.79], $d = 5.18$, $B_{H(0,1)} = 1.70$]. The same was true for heart rate variability (substantial evidence; $t(34) = -0.195$, $p = 0.846$, 95% CI [-21.2, 17.7], $d = 0.067$, $B_{H(0,1)} = 3.94$). For the asynchronous condition, there were also no group differences in heart rate (anecdotal evidence; $t(34) = -1.344$, $p = 0.188$, 95% CI [-13.6, 0.255], $d = 0.48$, $B_{H(0,1)} = 1.80$) nor heart rate variability (substantial evidence; $t(34) = 0.786$, $p = 0.437$, 95% CI [-9.88, 21.7], $d = 0.28$, $B_{H(0,1)} = 3.07$). Together, these results indicate with moderate evidence that cardiac physiology, encompassing 'sympathetic' heart rate and parasympathetic RMSSD, did not differentiate between individuals who did and did not experience the rubber hand illusion.

**Electrodermal activity differences between synchronous and asynchronous conditions.** We observed no significant difference between the synchronous and asynchronous conditions in the mean amplitude of skin conductance responses (substantial evidence; $t(36) = -0.988$, $p = 0.330$, 95% CI [-2.45, 12.92], $d = 0.206$, $B_{H(0,1)} = 4.88$); nor frequency (substantial evidence; $t(36) = -0.911$, $p = 0.369$, 95% CI [-2.45, 12.92], $d = 0.089$, $B_{H(0,1)} = 5.235$) over the two minute of both conditions. As with the cardiac measures, the data were collected over the two minutes during which the participant received regular somatosensory stimulation. f SCR events were infrequent (mean ± std synchronous condition 5.92 ± 7.3; asynchronous 6.65 ± 9.0).

**Electrodermal activity differences between participants who experienced the illusion and those who did not.** For the synchronous condition, we observed no differences in mean amplitude of skin conductance responses (SCRs) between participants who did and did not experience the rubber hand illusion (very strong evidence; $t(34) = -0.561$, $p = 0.578$, 95% CI [-0.76, 0.051], $d = 3.36$, $B_{H(0,1)} = 13.50$). The same was true for frequency of SCRs (substantial evidence; $t(34) = -0.373$, $p = 0.711$, 95% CI [-6.51 14.60], $d = 0.13$, $B_{H(0,1)} = 3.94$). For the asynchronous condition, there were also no observed group differences in mean amplitude of SCRs (substantial evidence; $t(34) = 0.60$, $p = 0.550$, 95% CI [-0.047, 0.094], $d = 0.22$, $B_{H(0,1)} =$

3.42), nor number of SCRs (substantial evidence t(34) = -0.21, p = 0.983, 95% CI [-6.17, 6.03], d = 0.01, $B_{H(0,1)}$ = 4.00). Together, these results indicate that sympathetic electrodermal activity (SCRs) did not differentiate the synchronous stroking condition, associated with the induction and experience of the rubber hand illusion from asynchronous 'illusion free' control condition. Moreover, SCRs did not differ between those individuals who experienced the illusion from those that did not.

**SKNA differences between synchronous and asynchronous conditions.** We tested for aSKNA effects at a slightly higher temporal resolution anticipating greater sensitivity [30]. Time-windows of 30s seconds were used to investigate differences between synchronous and asynchronous conditions. Four time-windows for each condition (30s, 60s, 90s, and 120s) were compared. The results obtained are as follows: 30s, t(36) = -1.311, p = 0.198, 95% CI [-0.02, 0.004], d = 0.011, $B_{H(0,1)}$ = 3.441; 60s, t(36) = -0.741, p = 0.464, 95% CI [-0.02, 0.009], d = -0.07, $B_{H(0,1)}$ = 5.991; 90s, t(36) = -0.674, p = 0.505, 95% CI [-0.01, 0.007], d = 0.05, $B_{H(0,1)}$ = 6.272; 120s, t(36) = 0.053, p = 0.958, 95% CI [-0.01, 0.01], d = 0.01, $B_{H(0,1)}$ = 7.809. Therefore, these data provide strong evidence for $H_0$, indicating that there are no differences in aSKNA between the two conditions (Fig 4).

**SKNA differences between participants who experienced the illusion and those who did not.** The aSKNA differences between participants who got the illusion and those who did not as indicated by the discriminant factor analysis were analysed for each condition and each time window. For the synchronous condition, the results are as follows: 30s, t(35) = -0.366, p = 0.716, 95% CI [-0.59, 0.40], d = 0.13, $B_{H(0,1)}$ = 3.777; 60s, t(35) = -0.368, p = 0.715, 95% CI [-0.59, 0.41], d = -0.13, $B_{H(0,1)}$ = 3.775; 90s, t(35) = -0.394, p = 0.696, 95% CI [-0.60, 0.40], d = 0.14, $B_{H(0,1)}$ = 3.743; 120s, t(35) = -0.358, p = 0.722, 95% CI [-0.59, 0.41], d = 0.13, $B_{H(0,1)}$ = 3.787. Therefore, there was substantial evidence for $H_0$, with Bayes factors higher than 3, indicating that there were no differences in aSKNA between the two groups in the synchronous condition.

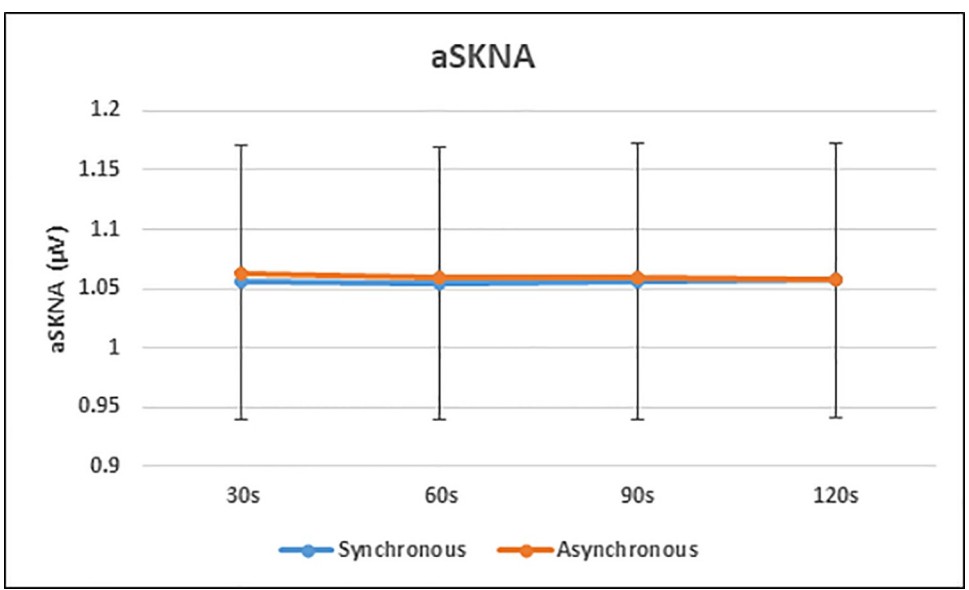

**Fig 4. Average skin nerve activity (aSKNA) for synchronous and asynchronous conditions over each 30 s time-window expressed as means ± 1SE.** Following established methods [30], continuous recording of SNKA was processed to give average scores over the 30s for each synchronous (active–associated with rubber hand illusion) and asynchronous (control–not associated with the rubber hand illusion) stroking conditions of the task. No consistent differences between conditions were observed across participants.

For the asynchronous condition, the results are as follows: 30s, $t(35) = -0.365$, $p = 0.717$, 95% CI [-0.60, 0.42], $d = 0.12$, $B_{H(0,1)} = 3.779$; 60s, $t(35) = -0.381$, $p = 0.705$, 95% CI [-0.61, 0.41], $d = 0.13$, $B_{H(0,1)} = 3.759$; 90s, $t(35) = -0.393$, $p = 0.697$, 95% CI [-0.61, 0.42], $d = 0.14$, $B_{H(0,1)} = 3.745$; 120s, $t(35) = -0.381$, $p = 0.705$, 95% CI [-0.61, 0.42], $d = 0.13$, $B_{H(0,1)} = 3.759$. Therefore, there was substantial evidence for $H_0$, with Bayes factors higher than 3, indicating that there were no differences in aSKNA between the two groups in the asynchronous condition (Fig 5).

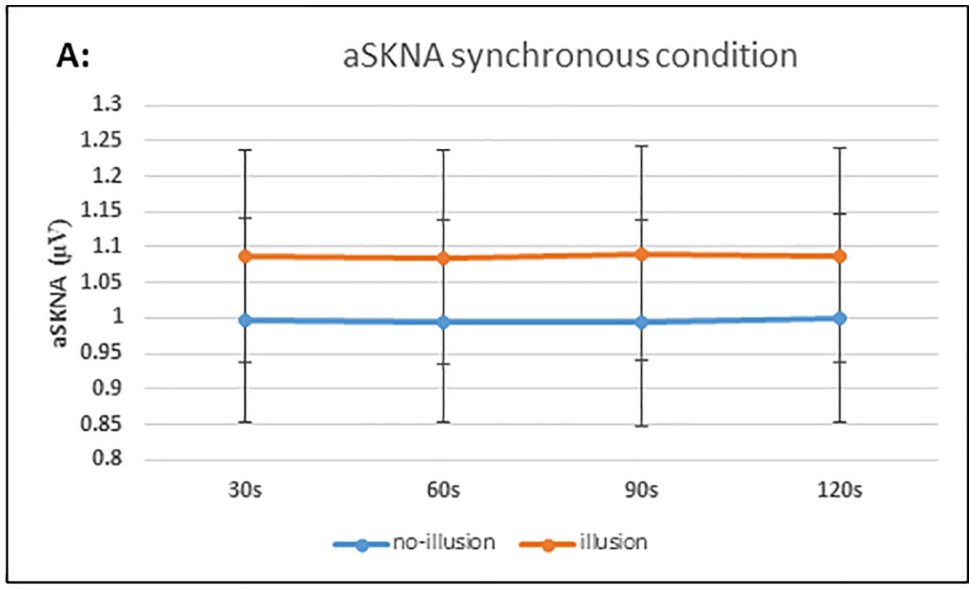

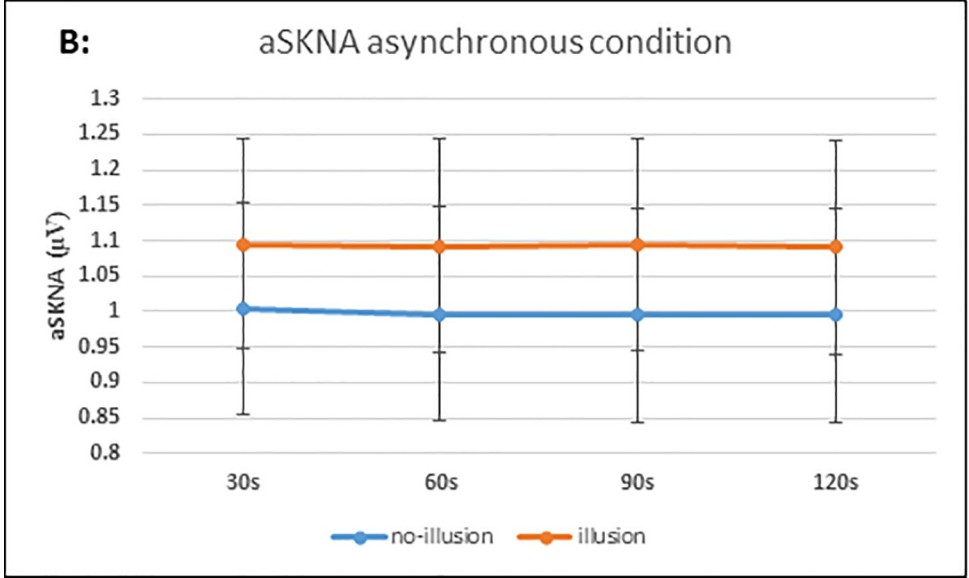

**Fig 5. Average skin nerve activity (aSKNA) in participants who were classified has having a strong versus minimal experience of the rubber hand illusion means ± 1SE.** Following the methods of Doytchinova and colleagues [30–32], continuous recording of SNKA was processed to give average scores over the 30s for each; A: Synchronous (active–associated with rubber hand illusion) and; B: Asynchronous (control–not associated with the rubber hand illusion) stroking conditions of the task. No differences between individual who did and did not experience the illusion were observed for either the active or control condition.

**Table 3. Correlations between Heart rate (HR), heart rate variability (HRV*), skin conductance responses (SCR) and aSKNA for synchronous (Syn) and asynchronous (Asyn) experimental conditions.**

|  | HR _Syn | HR _Asyn | HRV _Syn | HRV _Asyn | SCR _Syn | SCR _Asyn | SNKA _Syn |
|---|---|---|---|---|---|---|---|
| **HR** **_Asyn** | **r = 0.888** **B = 0.000** | | | | | | |
| HRV _Syn | R = -0.317 B = 1.278 | r = -0.139 B = 5.587 | | | | | |
| HRV _Asyn | **r = -0.640** **B = 0.001** | **r = -0.593** **B = 0.005** | **r = 0.605** **B = 0.003** | | | | |
| SCR _Syn | R = 0.094 B = 6.723 | r = 0.039 B = 7.618 | r = -0.110 B = 6.339 | r = -0.156 B = 5.136 | | | |
| SCR _Asyn | R = -0.040 B = 7.606 | r = -0.059 B = 7.366 | r = -0.251 B = 2.576 | r = -0.214 B = 3.508 | r = 0.220 B = 3.341 | | |
| SKNA _Syn | R = 0.033 B = 7.677 | r = 0.033 B = 7.677 | r = 0.258 B = 2.395 | r = 0.195 B = 4.023 | r = -0.192 B = 4.103 | r = -0.045 B = 7.555 | |
| SKNA _Asyn | r = 0.033 B = 7.679 | r = 0.035 B = 7.659 | r = 0.254 B = 2.497 | r = 0.190 B = 4.156 | r = -0.192 B = 4.102 | r = -0.052 B = 7.470 | **r = 0.999** **B = 0.000** |

*Root mean square of successive differences; RMSSD [39].

**Correlations across physiological measures.** We computed correlations across autonomic measures, heart rate, heart rate variability, and skin conductance responses, and aSKNA for both synchronous and asynchronous conditions (Table 3). Overall, these revealed little evidence between synchronous and asynchronous conditions (hence rubber hand illusion) for a systematic shift in the relationship between physiological variables that could represent a difference in patterning of bodily control. In particularly, we tested explicitly for a changing relationship between heart rate and heart rate variability. Typically, heart rate increases in heart rate are balanced by heart rate variability, reflecting baroreflex activity. In stress/arousal states suppression of the baroreflex may change this relationship by inhibiting cardiovagal tone, decreasing heart rate variability and allowing heart rate and blood pressure to rise unchecked. We observed that during the asynchronous (control) condition heart rate was significantly negatively correlated with heart rate variability (R(35) = -0.593; p = 0.000, 95CI [-0.769, -0.333] B = 0.005). In the sychronous condition, associated with experience of the rubber hand illusion, this relationship reduced in strength (R(35) -0.317, p = 0.056, 95CI [-0.581, 0.008] B = 1.278). However, the test of the interaction, i.e. differences between the HR-HRV correlation for synchronous versus asynchronous conditions, did to reach significance Z = -1.460, p = 0.144, indicating a lack of compelling evidence across participants of a change in cardiac as a consequence of the rubber hand illusion induction (or effect).

## Interoceptive accuracy

To test whether individual differences in interoceptive sensitivity influenced the experience of the rubber hand illusions in this sample [5], first a median split was used to divide participants into high interoceptive accuracy and low interoceptive accuracy groups [35]. We then tested for group differences between in strength of the rubber hand illusion. During synchronous proprioceptive drift, there was substantial evidence for no observable difference between high and low interceptive accuracy groups (t(32) = -0.734, p = 0.468, 95% CI [-15.48, 32.94], d = 0.25, $B_{H(0,1)}$ = 3.183). Similarly, there was substantial evidence for no difference for

asynchronous proprioceptive drift, arguably a control for 'suggestibility' [33] (t(32) = 0.466, p = 0.644, 95% CI [-12.56, 20.02], d = 0.16, $B_{H(0,1)}$ = 3.656).

Regarding subjective ratings in the synchronous condition, there were no differences on any of the subscales; ownership (t(32) = 1.132, p = 0.266, $B_{H(0,1)}$ = 2.323); location (t(32) = -0.344, p = 0.733, $B_{H(0,1)}$ = 3.816); agency (t(32) = -0.225, p = 0.824, $B_{H(0,1)}$ = 3.929). Regarding subjective ratings in the asynchronous condition, there were no differences on any of the sub-scales: ownership (t(32) = 0.983, p = 0.334, $B_{H(0,1)}$ = 2.252); location (t(32) = -0.249, p = 0.805, $B_{H(0,1)}$ = 3.910); agency (t(32) = -0.239, p = 0.813, $B_{H(0,1)}$ = 3.918). Again, for both synchronous and asynchronous conditions, Bayes factors showed the evidence for no interoceptive group effects to be substantial for location and agency while the measures for ownership was anec-dotal [40].

Correlation analyses were run between measures of proprioceptive drift and interoceptive accuracy and awareness scores. There was substantial evidence for no correlation between interoceptive accuracy and synchronous drift (r = -0.277, p = 0.197, $B_{H(0, 1)}$ = 3.284), asynchro-nous drift (r = -0.138, p = 0.436, $B_{H(0, 1)}$ = 5.561), nor the difference between synchronous and asynchronous drift (r = -0.107, p = 0.547, $B_{H(0, 1)}$ = 6.273). The same trend was observed for interoceptive awareness and synchronous drift (r = -0.064, p = 0.721, $B_{H(0, 1)}$ = 7.050), asyn-chronous drift (r = -0.097, p = 0.585, $B_{H(0, 1)}$ = 6.479), and the difference between synchronous and asynchronous drift (r = -0.001, p = 0.996, $B_{H(0, 1)}$ = 7.512). Together, these results indicate no significant difference between low- and high-interoceptive perceivers in the strength of the illusion as indicated by proprioceptive drift and/or subjective ratings.

## Discussion

We set out to test whether the induction and experience of the rubber hand illusion is associ-ated with reliable embodied changes in peripheral systemic (rather than localised) autonomic function, linked theoretically to the notion of interoceptive predictive coding as a basis to the integrity of conscious self-representation. The initial ambition was to extract, from measures of peripheral physiology, a change in efferent autonomic drive [6–10,19,25,31,41]. This, we hypothesised, might encode a change in interoceptive prediction as a signature of the shift in the conscious experience of bodily self when an artificial limb is adopted as part of one's own body [11–17]. Our study incorporated two important features: first, the use of SKNA, a novel approach to record non-invasively from sympathetic nerves [30–32] and; second, the use of Bayes factors to determine strength of evidence for and against the null hypothesis [40]. In our findings, we produced robust evidence for successful induction of the rubber hand illusion, yet we observed no systematic changes across cardiac and electrodermal autonomic measures dur-ing the induction procedure (synchronous stroking) relative to a control procedure (asynchro-nous stroking). Moreover, when participants were partitioned by a discriminant analysis according to whether or not the illusory experience was strong, autonomic measures in either condition did not distinguish between these two groups. We also did not replicate a previously reported association between sensitivity to internal bodily signals (as indexed heuristically by performance accuracy on a heartbeat tracking task) and lower susceptibility to the rubber hand illusion [5].

Our results, in keeping with previously work, showed a partial correlation between proprio-ceptive drift and subjective ratings, re-confirming previous evidence to suggest that proprio-ceptive drift and subjective ratings are two independent measures of the rubber hand illusion [37,42,43], but see also [36]. Furthermore, these measures can reflect different dimensions of bodily ownership: body-location (linked to proprioceptive drift) and body-ownership (linked to the subjective experience of ownership and/or agency) [44]. Our overarching hypothesis

predicted that the subjective representation of bodily self and agency might have the closest link to the regulation in internal interoceptive state.

Retrospectively, seeking a bodily signature of the rubber hand illusion was ambitious. We pursued more systemic autonomic measures, expressed thoughout the body (cardiovascular, electrodermal and SKNA responses), rather than more limb-specific change. Previous reports of hand cooling associated with rubber hand illusion found this to be localised to the specific side of the illusion [19].

We measured heart rate and heart rate variability, both indices of central autonomic cardiac control with systemic impact, capturing the interaction of sympathetic and parasympathetic (vagus nerve) cardiac innervation [38,39]. Heart rate itself is often used as a proxy for sympathetic cardiac influences; heart rate typically increases with decreasing heart rate variability as changes in sympathovagal balance produce cardiovascular arousal. Heart rate variability in particularly is a widely used measure linked to health, well-being and cognitive flexibility, where decreases in heart rate variability accompany stress, anxiety and negative affective experience [45,46]. These decreases typically relate to a suppression of the baroreflex by top-down brain signals, enabling heart rate and blood pressure to rise together to meet behavioural allostatic demands. However, within our data, we did not find any clear association with the experience of illusory limb ownership, suggesting cardiovascular homeostasis need not adapt to this change in body representation.

Our findings of no sympathetic electrodermal or SKNA changes associated with the rubber hand illusion contrast with recent observations that differences in skin conductance occur during the rubber hand illusion [25]. Yet even in this study, transient effects were observed, suggesting the novelty of the experience was a contributing factor. Increases in phasic and tonic electrodermal activity were also reported in an earlier study, associated with the period leading up to the onset of the illusory experience of body ownership, and amplified in people who report high levels of anomalous bodily experiences [41]. Both these studies suggest that autonomic changes measurable electrodermally, if they occur, may be brief, and mostly related to the cognitive and emotional and somatosensory experience of the induction process.

Those rubber hand illusion studies that report a fall in skin temperature in the 'replaced hand' have been influential and widely cited as evidence for a deep physical embodiment [19]. Temperature after a 7–8 min stroking period is reported as lower during synchronous compared to asynchronous stroking on the test hand. However, no temperature difference is reported on the non-stimulated hand. This observation suggests that hand cooling might relate to the illusionary disowning of the real hand in favour of the rubber substitute. Alternative explanations include greater immobility of the target hand (during synchronous stroking) and/or the illusion) features of the synchronous stoking that might better elicit peripheral e.g. axo-axonal reflex hypo-perfusion. Social factors may also play a role [22]. Mechanisms for observed hand cooling are relevant to other observations: While passive disowning of the replaced hand most likely is accompanied by reduced autonomic nervous drive to the limb, the increase in electrodermal activity occasionally reported with the rubber hand illusion is also consistent with activation of sympathetic vasoconstrictor neurons within skin vasculature, which can actively reduce skin temperature [47]. However, the effect appear unreliable, with studies not consistently observing cooling associated with the rubber hand illusion [21]. In our study, there was strong evidence for no effect of the procedure or the experience of the illusion on either electrodermal activity in the hand or skin sympathetic activity measured from the chest surface. Nevertheless, our two-minute induction process was relatively short and it remains possible that transient changes at the onset of the illusion were not sustained long enough to affect the average skin conductance data. For SKNA measures, although we analysed these reportedly sensitive response data over shorter (30 s) time-periods over the course

of each experimental condition, we nevertheless identified no effect to suggest a transient response.

Sympathetic nerve traffic can be quantified using microneurography [29,48–50], in which microelectrodes are inserted into peripheral nerves (median or common peroneal) to record from sympathetic axon bundles [29,48–52]. The relatively recent introduction of non-invasive skin sympathetic nerve activity (SKNA) recordings provides a means of overcoming some of the logistical limitations of invasive microneurography [30–32,51]. Subcutaneous nerve activity recorded from the chest wall of dogs correlates with nerve activity within stellate ganglion and corresponding sympathetic nervous effects on the heart [52]. This SKNA is reportedly a more sensitive index of cardiac sympathetic tone than heart rate variability derived measures [53]. Validation studies in humans have established the sensitivity of SKNA recordings to autonomic challenges in healthy individuals, to lidocaine deactivation of stellate ganglion, to vagus nerve stimulation, and to pathological changes in cardiac rhythm [30,32,51]. As a measure of multiunit sympathetic activity relevant to cardiovascular control, the research potential of this approach is exciting. Innovatively, our experimental study applied published methods for recording SKNA [30,32] to test for efferent neural signatures of central changes in bodily self-representation. Contrary to our predictions however, we report no differences in this purportedly sensitive measure between the active (synchronous stroking) and control (asynchronous stroking) conditions of the rubber hand illusion, whether or not a strong experience of change in embodied representation occurred.

Here we examined the rubber hand illusion, rather than the arguably more encompassing experience of a full body illusion, typically achieved using virtual reality head mounted displays [18,54,55]. The full body illusion affects the whole individual and the strength of body ownership can be measured through whole body spatial drift (i.e. shift in perceived location) as well as through subjective ratings. In comparison, the local modulation of body ownership with induction of the rubber hand illusion represents a limitation of the present study. Where tonic changes in physiological state are reported in association with the rubber hand illusion (e.g. temperature drop), such changes seem only to occur in the disowned arm [19,20]. In contrast, some studies of the full body illusion report widespread reductions in body temperature [54]. Moreover, visual feedback of 'systemic' heartbeat rhythms feedback will enhance the experience of full body illusions [18,55,56] (and the rubber hand illusion [17]). However, and perhaps contrary to what we may have predicted, studies of the full body illusion have failed to demonstrate significant tonic changes in autonomic state including heart rate and heart rate variability [18], and skin conductance level SCL [57]. This was even the case for studies in which cardiovisual synchrony enhanced the experience of the full body illusion [18,56], even though cardiac autonomic entrainment might facilitate the illusory experience. However, as with the rubber hand illusion [6,8], phasic electrodermal response (SCR) evoked by contact and reaction to pain are affected by illusory body ownership [57,58]. Such altered autonomic reactivity (even in the absence of a tonic autonomic change) nevertheless suggests that the full body illusion is accompanied by a resetting of central physiological regulatory sensitivity. One study of the full body illusion [59] observed changes in electrocortical rhythms during illusory self-location, notably differences in alpha power over sensorimotor and precentral cortical regions: While both brain alpha oscillations and autonomic (electrodermal or cardiac) measures are both modulated by attention and cognitive arousal, there is no simple interdependence between these central and peripheral measures. However, the identified cortical brain regions are known sources of autonomic drive (e.g. in motor central command) [60], suggesting that the electrocortical change may affect the regulation and reactivity of bodily physiology (even interoceptive prediction) in context of the whole body illusion. It remains unclear if the

experience of the full body illusion would be better reflected in direct measures of sympathetic nerve traffic (SKNA).

One interesting challenge within the field of body-brain Interaction (including illusions of body ownership, interoception and autonomic psychophysiology) is the pursuit of objective measures, unbiased by situational and individual confounds such implicit task demands, response bias, and suggestibility. Tests of interoceptive sensitivity, especially the heartbeat tracking task, have been criticised for not providing an unconfounded measure of sensory perception; individual differences in performance are subject to a number of non-sensory factors including knowledge of one's heart rate practice effects and other top down beliefs [61,62]. These widely acknowledged associations are of lesser or greater relevance depending on what question the research study aims to address and, correspondingly, how the data are interpreted [62]. Relevant to this study, individual differences in social compliance, suggestibility and/or neuroticism are recognized determinants of behaviours relating to 'strength of self-representation'. While clinic drug trials are 'blinded' to manage the unwanted influence of expectancies and biases, suggestibility (including hypnotizability/phenomenological control) influence many tasks designed to access conscious processes, including the rubber hand illusion [63]. Individual differences in suggestibility can account for around 10 percent of the variance in subjective measures of the rubber hand illusion [63]. Such observations indicate that superordinate domain-general beliefs and expectancies influence perceptual behavioural and experiential measures. Moreover, they highlight the importance of participant-level susceptibility and experiment-level context in shaping subjective experience. In this study, we did not assess if suggestibility influenced autonomic responses to the induction or experience of the rubber hand illusion, beyond using the established asynchronous stroking as a standard control condition. Our autonomic data showed a lack of difference between synchronous and asynchronous conditions and between participants who did and did not experience the illusion. Our findings do not test whether suggestion affects autonomic correlates of the rubber hand illusion (as is observed with visual imagery [64]). Moreover, we also did not observe effects associated with individual differences in performance of a heartbeat counting task, a heuristic measure of interoceptive perception that is recognised to be sensitive to top-down expectancies and bias [61,62]. Increased performance on this task is linked to enhanced autonomic reactivity and to resistance to rubber hand illusion [5]. Although this latter observation has not always been replicated [23,24], it remains somewhat paradoxical that suggestibility facilitates both heartbeat tracking performance measures and experience of the rubber hand illusion.

In conclusion, we undertook detailed physiological monitoring of participants during the rubber hand illusion. We found no systematic effect on state measures of autonomic activity that might discriminate between the induction procedure, or the experience of owning an illusory body part when compared to a control condition. We obtained, for the first time in this context, SKNA recordings, indexing skin sympathetic nerve activity proximally related to stellate ganglion activity. We used Bayes factors to confirm the absence of autonomic differences. Our study illustrates that tonic changes in bodily physiology are not obligatory accompaniments of an alteration in this type of conscious self-represen**t**ation.

## Supporting information

**S1 File.**
(DAT)

**S2 File.**
(DOCX)

## Acknowledgments

We are grateful for the advice and help of Dr David Watson in setting up the data acquisition platform, and to Prof Peter Taggart for highlighting and advising on the SKNA methodology.

## Author Contributions

**Conceptualization:** Hugo D. Critchley, Jamie Ward.

**Data curation:** Vanessa Botan.

**Formal analysis:** Hugo D. Critchley, Vanessa Botan.

**Funding acquisition:** Hugo D. Critchley, Jamie Ward.

**Investigation:** Vanessa Botan, Jamie Ward.

**Methodology:** Hugo D. Critchley, Vanessa Botan, Jamie Ward.

**Project administration:** Hugo D. Critchley, Vanessa Botan, Jamie Ward.

**Resources:** Hugo D. Critchley, Jamie Ward.

**Supervision:** Hugo D. Critchley, Jamie Ward.

**Validation:** Jamie Ward.

**Writing – original draft:** Hugo D. Critchley, Vanessa Botan, Jamie Ward.

**Writing – review & editing:** Hugo D. Critchley, Vanessa Botan, Jamie Ward.

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
