## [Decision Letter · Decision Letter 0]

20 Nov 2020

PONE-D-20-22568

Absence of  reliable  physiological signature of  illusory body ownership  revealed by f ine-grained autonomic measurement during the rubber hand illusion

PLOS ONE

Dear Dr. Critchley,

Thank you for submitting your manuscript to PLOS ONE. I am very sorry for the long time it has taken to review your paper. We have had enormous difficulties in securing expert reviews but I'm pleased to inform you that we have just received two reviews back. Both reviewers were positive about the importance and quality of your study and have only minor revisions to recommend. We invite you to submit a revised version of the manuscript that addresses the points raised during the review process.

We look forward to receiving your revised manuscript.

Kind regards,

Jane Elizabeth Aspell, PhD

Academic Editor

PLOS ONE

Journal Requirements:

Reviewers' comments:

Reviewer's Responses to Questions

**Comments to the Author**

1. Is the manuscript technically sound, and do the data support the conclusions?

Reviewer #1: Yes

Reviewer #2: Yes

2. Has the statistical analysis been performed appropriately and rigorously? 

Reviewer #1: Yes

Reviewer #2: Yes

3. Have the authors made all data underlying the findings in their manuscript fully available?

Reviewer #1: Yes

Reviewer #2: No

4. Is the manuscript presented in an intelligible fashion and written in standard English?

Reviewer #1: Yes

Reviewer #2: Yes

5. Review Comments to the Author

Reviewer #1: I support the publication of negative results, and the manuscript in its current from only needs minor revisions

- in the revised manuscript please provide line numbers

- please include a sample size calculation

- referring to interoceptive predictive coding the authors might include the work of Lisa Feldmann Barrett and W. K. Simmons

- maybe the authors could discuss if for example a full body illusion (through visuo-tactile stimulation or cardio-visual stimulation) would have had a different effect on the HRV and the SKNA (see work by Lenggenhager et al, Heydrich and Aspell et al., Romano et al, Behav Brain Res, 2014)

- the procedure of splitting the sample in two groups of high and low interoceptive accuracy is not clear to me: how big are the two groups? what was the median? why is the degree of freedom suddenly t(32) comparing the two groups?

- in the discussion there are two samll typos: page 21, line 4: whereas should be removed and page 22, last paragraph: individual difference IN performance...

- in general I think the discussion is a bit too general (whole discussion on the reliability of the SKNA) but not really focussed on the actual study (maybe the discussion could be even shortened or focussed on other aspects, see point on full body illusion etc)

Reviewer #2: The authors use a rubber hand illusion paradigm to measure the relationship between transient alterations in hand ownership and changes in various parameters of physiological activity. Rather disappointing, they do not find any modulation of these parameters, despite successfully inducing the RHI according to both questionnaire and proprioceptive drift measures. In my opinion this is a very important study, which might contribute to the increasing but often contradicting results on the very interesting relationship between interoceptive and exteroceptive body perception. The paper is very clearly written, the experimental setup and the statistical analysis in my opinion sound, the results complete and interesting (even if null results) and the discussion sound. I only have a few rather minor comments (see below).

Abstract:

• Some typos, e.g. influential

Introduction:

• While the authors partially mention the problem of replication for some of the described effects (e.g. for temperature drop), they should include these also for others of the measure (e.g. response to threat).

• A hypothesis for the interoceptive awareness should be given, or else it should be more clearly labeled as explorative analysis

Methods:

• A sample size justification is missing

• Handedness should be reported (if assessed)

• In the statistical analysis the authors do not report the within comparison of condition on the questionnaire and drift (only the analysis to re-confirm). Generally, I think this section could profit from some more clarity, e.g. by restructuring the statistics according to the hypothesis (and in the same order)

• The information about the median split for good versus bad perceivers should be added in the statistical analysis part already

• For the subjective reports, I imagine the scale is from 1-7? It might be good to plot the full scale, as for example for agency, participants in general rather disagreed, which is not really clear the way it is plotted currently.

• It is mentioned in the section on the discriminative analysis that the two predictors (proprioceptive drift and illusion score) are not strongly correlated, which is good for the analysis, but conceptually not really. It might be good to mention/discuss this result and link it to findings that suggest no or weak correlation between different measures of the illusion.

Results:

• Plot of proprioceptive drift: I suggest plotting median and CI instead of mean and standard error for more informative plotting and PD should be mentioned in the legend

• Why was the SKNA analysis done separately for the 30s intervals but not the other physiological measures?

Discussion:

• Reference to the previously reported correlation between interoceptive measures and embodiment is lacking in the first paragraph.

• Generally, often claims are made without referencing, (e.g. studies have criticized measures of interception – I would suggest a few references to such claims

• I think it might be interesting, discussing a bit more the fact that the authors used a very local modulation of bodily self consciousness (illusory ownership of a hand), which might be a limitation for this more general homeostatic process and also the heart-related measures. Wouldn’t you expect these measures to be more influenced by full body illusions or illusions in VR?

6. PLOS authors have the option to publish the peer review history of their article (what does this mean?). If published, this will include your full peer review and any attached files.

Reviewer #1: No

Reviewer #2: No

---

## [Author Response · Author response to Decision Letter 0]

12 Feb 2021

Dear Dr Aspell,

Re manuscript:

Absence of reliable physiological signature of illusory body ownership revealed by fine-grained autonomic measurement during the rubber hand illusion

Authors: ` Hugo D Critchley, Vanessa Botan, Jamie Ward 

Please may we submit our full revision of this manuscript, which we have amended in response to the helpful and insightful comments of the two Reviewers. Below we list the response to the specific issues raised by each of the reviewers and describe how we have addressed these. We consider that our manuscript has improved as a consequence of these changes and hope that it well be viewed as now ready for publication in Plos One.

Sincerely

Hugo Critchley on behalf of all authors 

Response to Reviewer 1

We are very grateful for the encouraging comments and helpful advice from Reviewer 1 We have responded to the Reviewer’s comments as detailed below, which have strengthened the manuscript. 

1) I support the publication of negative results, and the manuscript in its current from only needs minor revisions

We are very grateful for the support of the Reviewer for our work

2) - in the revised manuscript please provide line numbers

We now provide line numbers in the revised manuscript

3) - please include a sample size calculation

Thank you for highlighting this omission:

We now included the following text within the methods section (lines 168-172) with associated references:

Sample size was computed using G-power calculator for paired t-tests. The computation parameters included a medium effect size (d=0.5), α set at 0.05, and power set at 0.8. The resulting sample size was of 34 participants. Considering that the effect sizes between synchronous and asynchronous conditions are usually medium/large [19,36,37] a sample of 34 participants was considered adequate.

4) - referring to interoceptive predictive coding the authors might include the work of Lisa Feldmann Barrett and W. K. Simmons

We now reference [4] the review paper: Barrett LF, Simmons WK. Interoceptive predictions in the brain. Nature Reviews Neuroscience 2015; 16: 419-29. 

5) maybe the authors could discuss if for example a full body illusion (through visuo-tactile stimulation or cardio-visual stimulation) would have had a different effect on the HRV and the SKNA (see work by Lenggenhager et al, Heydrich and Aspell et al., Romano et al, Behav Brain Res, 2014)

Thank you for this very helpful guidance. Yes we now discuss to the full body illusion (quoting the empirical studies highlighted by the Reviewer, which do include autonomic measures. We refer to this in the abstract and now include the following information within our revision of the Discussion (lines 684-742).

Here we examined the rubber hand illusion, rather than the arguably more encompassing experience of a full body illusion, typically achieved using virtual reality head mounted displays [18, 56, 57]. The full body illusion affects the whole individual and the strength of body ownership can be measured through whole body spatial drift (i.e. shift in perceived location) as well as through subjective ratings. In comparison, the local modulation of body ownership with induction of the rubber hand illusion represents a limitation of the present study. Where tonic changes in physiological state are reported in association with the rubber hand illusion (e.g. temperature drop), such changes seem only to occur in the disowned arm [19, 20]. In contrast, some studies of the full body illusion report widespread reductions in body temperature [56]. Moreover, visual feedback of ‘systemic’ heartbeat rhythms feedback will enhance the experience of full body illusions [18, 57, 58] (and the rubber hand illusion [13]). However, and perhaps contrary to what we may have predicted, studies of the full body illusion have failed to demonstrate significant tonic changes in autonomic state including heart rate and heart rate variability [18], and skin conductance level SCL [59]. This was even the case for studies in which cardiovisual synchrony enhanced the experience of the full body illusion [18, 57], even though cardiac autonomic entrainment might facilitate the illusory experience. However, as with the rubber hand illusion [6, 8], phasic electrodermal response (SCR) evoked by contact and reaction to pain are affected by illusory body ownership [59, 60]. Such altered autonomic reactivity (even in the absence of a tonic autonomic change) nevertheless suggests that the full body illusion is accompanied by a resetting of central physiological regulatory sensitivity. One study of the full body illusion [61] observed changes in electrocortical rhythms during illusory self-location, notably differences in alpha power over sensorimotor and precentral cortical regions: While both brain alpha oscillations and autonomic (electrodermal or cardiac) measures are both modulated by attention and cognitive arousal, there is no simple interdependence between these central and peripheral measures. However, the identified cortical brain regions are known sources of autonomic drive (e.g. in motor central command) [62], suggesting that the electrocortical change may affect the regulation and reactivity of bodily physiology (even interoceptive prediction) in context of the whole body illusion. It remains unclear if the experience of the full body illusion would be better reflected in direct measures of sympathetic nerve traffic (SKNA).

6) - the procedure of splitting the sample in two groups of high and low interoceptive accuracy is not clear to me: how big are the two groups? what was the median? why is the degree of freedom suddenly t(32) comparing the two groups?

We now describe this explicitly within the text Methods section, lines 293-297:

A median split was performed in exploratory analysis of the relationship between interoceptive performance on the heartbeat tracking task, The two groups (low and high interoception) had 17 participants each and the median point was 0.71. Due to technical issues, interoceptive data was not available for three participants (16 in the higher interoception group, 14 in the lower interoception group). Therefore, this reduced the degrees of freedom to 32.

7) - in the discussion there are two small typos: page 21, line 4: whereas should be removed and page 22, last paragraph: individual difference IN performance...

Thank you. We have corrected these typos and checked the paper again to remove other similar errors

8) - in general I think the discussion is a bit too general (whole discussion on the reliability of the SKNA) but not really focussed on the actual study (maybe the discussion could be even shortened or focussed on other aspects, see point on full body illusion etc).

We have more focussed more on the discussion of the relevant findings other aspects including the full body illusion (as described above). We have reduced the discussion relating to SKNA but consider some discussion is needed given the relative novelty of its application here. 

 

Response to Reviewer 2

We are very grateful for the helpful and supportive comments of Reviewer 2. The changes that we have made to address the reviewer’s points have improved the manuscript. We list our responses below:

1) The authors use a rubber hand illusion paradigm to measure the relationship between transient alterations in hand ownership and changes in various parameters of physiological activity. Rather disappointing, they do not find any modulation of these parameters, despite successfully inducing the RHI according to both questionnaire and proprioceptive drift measures. In my opinion this is a very important study, which might contribute to the increasing but often contradicting results on the very interesting relationship between interoceptive and exteroceptive body perception. The paper is very clearly written, the experimental setup and the statistical analysis in my opinion sound, the results complete and interesting (even if null results) and the discussion sound. I only have a few rather minor comments (see below).

Thank you for the encouragement to report our findings, which we consider to be important for a broad field encompassing the science of consciousness, self-representation and autonomic psychophysiology. We anticipate that the paper will be of interest to a wide readership and hope our findings will serve to guide future studies. 

2) Abstract: Some typos, e.g. influential

Thank you: We have endeavoured to ensure no further typos break through into the revised paper.

3) While the authors partially mention the problem of replication for some of the described effects (e.g. for temperature drop), they should include these also for others of the measure (e.g. response to threat).

Thank you for this. Yes, we have been focusing on issues around tonic autonomic change and less on evoked responses e.g. to threat which we had assumed to be more robust without detailed examination of this aspect of the literature. We now include the following line in the introduction (lines 54-57):

However, even despite the relative consistency across studies of exaggerated reactivity to threats to an embodied rubber hand [8], these effects may be more nuanced, in some cases relating more to physical contact than threat [9,10].

4) A hypothesis for the interoceptive awareness should be given, or else it should be more clearly labeled as explorative analysis

Although heartbeat tracking task performance and its association with relative resistance to the rubber hand illusion has been examine, there is little information regarding the association between interoceptive metacognitive awareness measures and the rubber hand illusion. Nevertheless, possible association with other perturbations of consciousness (e.g. dissociative symptoms) led us to explore if interoceptive awareness did affect experiential or autonomic correlates of the rubber hand illusion. We had not powered the study to test a prediction that poor performance and metacognition would enhance susceptibility to the illusion. We make it clear that our analyses relating to interoceptive measures of performance and (metacognitive) awareness are exploratory (lines 150-155).

Motivated by the theoretical and empirical link to perceptual aspects of interoception [5, 13, 14, 17, 18], we undertook exploratory analyses to test for associations between (objective and subjective) measures of the rubber hand illusion with individual differences in heartbeat counting performance and interoceptive metacognitive insight (awareness) [34,35].

5) Methods: A sample size justification is missing

Apologies for this omission this is now addressed within the methods section (lines 168-172) with associated references:

Sample size was computed using G-power calculator for paired t-tests. The computation parameters included a medium effect size (d=0.5), α set at 0.05, and power set at 0.8. The resulting sample size was of 34 participants. Considering that the effect sizes between synchronous and asynchronous conditions are usually medium/large [19,36,37] a sample of 34 participants was considered adequate.

6) Handedness should be reported (if assessed)

This was an oversight: Handedness is not formally recorded in the compiled database.

7) In the statistical analysis the authors do not report the within comparison of condition on the questionnaire and drift (only the analysis to re-confirm). Generally, I think this section could profit from some more clarity, e.g. by restructuring the statistics according to the hypothesis (and in the same order)

Thank you for pointing this out. We have restructured the presentation of statistics to maintain the same logical order throughout the manuscript. This has improved readability.

8) The information about the median split for good versus bad perceivers should be added in the statistical analysis part already

We have now added this information in the Methods section, lines 293-297:

A median split was performed in exploratory analysis of the relationship between interoceptive performance on the heartbeat tracking task, The two groups (low and high interoception) had 17 participants each and the median point was 0.71. Due to technical issues, interoceptive data was not available for three participants (16 in the higher interoception group, 14 in the lower interoception group). Therefore, this reduced the degrees of freedom to 32.

9) For the subjective reports, I imagine the scale is from 1-7? It might be good to plot the full scale, as for example for agency, participants in general rather disagreed, which is not really clear the way it is plotted currently.

Yes, this is correct. The scale from 1-7 s described in the Methods section, lines 197-200. And is also referred to in Table 2’

After each condition, the participant completed the RHI questionnaire comprising 10 items divided into three subscales: ownership, location, and agency, see Table 1 for further details. The items were measured on a 7-point Likert scale (1=strongly disagree, 7=strongly agree) and the average score for each subscale was calculated [33].

We now include (Fig 3) plots of participant ratings in relation to the subjective scale in the Results section 

Figure 3. Distribution of subjective ratings for each condition and each subscale. 

10) It is mentioned in the section on the discriminative analysis that the two predictors (proprioceptive drift and illusion score) are not strongly correlated, which is good for the analysis, but conceptually not really. It might be good to mention/discuss this result and link it to findings that suggest no or weak correlation between different measures of the illusion.

Thank you for raising this point. We have added the following section to the discussion; Line xx

Our results, in keeping with previously work, showed a partial correlation between proprioceptive drift and subjective ratings, re-confirming previous evidence to suggest that proprioceptive drift and subjective ratings are two independent measures of the rubber hand illusion [37, 43, 44] [but see also 36]. Furthermore, these measures can reflect different dimensions of bodily ownership: body-location (linked to proprioceptive drift) and body-ownership (linked to the subjective experience of ownership and/or agency) [45]. Our overarching hypothesis predicted that the subjective representation of bodily self and agency might have the closest link to the regulation in internal interoceptive state. 

11) Results: Plot of proprioceptive drift: I suggest plotting median and CI instead of mean and standard error for more informative plotting and PD should be mentioned in the legend.

Thank you for this helpful advice. We now include the following graph within our revised manuscript 

Figure 2. Proprioceptive Drift in Synchronous and Asynchronous conditions expressed in millimetres (mm: Median ± 95% CI).

12) Why was the SKNA analysis done separately for the 30s intervals but not the other physiological measures? 

We had focused much of the paper on SKNA, on the basis that this approach may be a more ‘fine grained’ index of autonomic change (relative to time required for reliable heart rate variability data, but also to measures of skin conductance response and heart rate). Measures of heart rate variability and frequency of SCRs (mean only ~6 over the 2 minutes) benefit in terms of reliability from longer sampling. We therefore tested specifically for more transient effects on SKNA, while also presenting the overall data for all measures, including average SKNA, in Table 3 (having established no evidence for shorter term modulation). We clarify this now in the Methods and Results.

13) Discussion: Reference to the previously reported correlation between interoceptive measures and embodiment is lacking in the first paragraph.

We have revised this first paragraph to reference this work, though the topic is elaborated more in the second paragraph.

14) Generally, often claims are made without referencing, (e.g. studies have criticized measures of interception – I would suggest a few references to such claims

We have added appropriate references to support such statements. e.g. regarding heartbeat tracking task (line 718-721)

15) I think it might be interesting, discussing a bit more the fact that the authors used a very local modulation of bodily self-consciousness (illusory ownership of a hand), which might be a limitation for this more general homeostatic process and also the heart-related measures. Wouldn’t you expect these measures to be more influenced by full body illusions or illusions in VR?

16) 

Thank you. Yes, we have now included a more extended discussion of this point as also requested by Reviewer 1 (lines 684-742).

Here we examined the rubber hand illusion, rather than the arguably more encompassing experience of a full body illusion, typically achieved using virtual reality head mounted displays [18, 56, 57]. The full body illusion affects the whole individual and the strength of body ownership can be measured through whole body spatial drift (i.e. shift in perceived location) as well as through subjective ratings. In comparison, the local modulation of body ownership with induction of the rubber hand illusion represents a limitation of the present study. Where tonic changes in physiological state are reported in association with the rubber hand illusion (e.g. temperature drop), such changes seem only to occur in the disowned arm [19, 20]. In contrast, some studies of the full body illusion report widespread reductions in body temperature [56]. Moreover, visual feedback of ‘systemic’ heartbeat rhythms feedback will enhance the experience of full body illusions [18, 57, 58] (and the rubber hand illusion [13]). However, and perhaps contrary to what we may have predicted, studies of the full body illusion have failed to demonstrate significant tonic changes in autonomic state including heart rate and heart rate variability [18], and skin conductance level SCL [59]. This was even the case for studies in which cardiovisual synchrony enhanced the experience of the full body illusion [18, 57], even though cardiac autonomic entrainment might facilitate the illusory experience. However, as with the rubber hand illusion [6, 8], phasic electrodermal response (SCR) evoked by contact and reaction to pain are affected by illusory body ownership [59, 60]. Such altered autonomic reactivity (even in the absence of a tonic autonomic change) nevertheless suggests that the full body illusion is accompanied by a resetting of central physiological regulatory sensitivity. One study of the full body illusion [61] observed changes in electrocortical rhythms during illusory self-location, notably differences in alpha power over sensorimotor and precentral cortical regions: While both brain alpha oscillations and autonomic (electrodermal or cardiac) measures are both modulated by attention and cognitive arousal, there is no simple interdependence between these central and peripheral measures. However, the identified cortical brain regions are known sources of autonomic drive (e.g. in motor central command) [62], suggesting that the electrocortical change may affect the regulation and reactivity of bodily physiology (even interoceptive prediction) in context of the whole body illusion. It remains unclear if the experience of the full body illusion would be better reflected in direct measures of sympathetic nerve traffic (SKNA).

---

## [Decision Letter · Decision Letter 1]

11 Mar 2021

Absence of  reliable  physiological signature of  illusory body ownership  revealed by fine-grained autonomic measurement during the rubber hand illusion

PONE-D-20-22568R1

Dear Dr. Critchley,

We’re pleased to inform you that your manuscript has been judged scientifically suitable for publication and will be formally accepted for publication once it meets all outstanding technical requirements.

Kind regards,

Jane Elizabeth Aspell, PhD

Academic Editor

PLOS ONE

Reviewers' comments:

Reviewer's Responses to Questions

**Comments to the Author**

1. If the authors have adequately addressed your comments raised in a previous round of review and you feel that this manuscript is now acceptable for publication, you may indicate that here to bypass the “Comments to the Author” section, enter your conflict of interest statement in the “Confidential to Editor” section, and submit your "Accept" recommendation.

Reviewer #1: All comments have been addressed

Reviewer #2: All comments have been addressed

2. Is the manuscript technically sound, and do the data support the conclusions?

Reviewer #1: Yes

Reviewer #2: Yes

3. Has the statistical analysis been performed appropriately and rigorously? 

Reviewer #1: Yes

Reviewer #2: Yes

4. Have the authors made all data underlying the findings in their manuscript fully available?

Reviewer #1: Yes

Reviewer #2: Yes

5. Is the manuscript presented in an intelligible fashion and written in standard English?

Reviewer #1: Yes

Reviewer #2: Yes

6. Review Comments to the Author

Reviewer #1: (No Response)

Reviewer #2: All comments have been adequately addressed and I suggest acceptance of the manuscript in the current format.

7. PLOS authors have the option to publish the peer review history of their article (what does this mean?). If published, this will include your full peer review and any attached files.

Reviewer #1: **Yes: **Lukas Heydrich

Reviewer #2: No

---

## [Editor Report · Acceptance letter]

16 Mar 2021

PONE-D-20-22568R1 

Absence of reliable physiological signature of illusory body ownership revealed by fine-grained autonomic measurement during the rubber hand illusion 

Dear Dr. Critchley:

I'm pleased to inform you that your manuscript has been deemed suitable for publication in PLOS ONE. Congratulations! Your manuscript is now with our production department. 

Kind regards, 

on behalf of

Dr. Jane Elizabeth Aspell 

Academic Editor

PLOS ONE